# Large Reductions in Satellite-Derived and Modelled European Lower Tropospheric Ozone During and After the COVID-19 Pandemic (2020–2022)

**Matilda. A. Pimlott[1], Richard. J. Pope[1,2], Brian. J. Kerridge[3,4], Richard. Siddans[3,4], Barry. G. Latter[3,4], Lucy. J. Ventress[3,4], Wuhu. Feng[1,5], Martyn. P. Chipperfield[1,2]**

[1]School of Earth and Environment, University of Leeds, Leeds, LS2 9JT, UK

[2]National Centre for Earth Observation, University of Leeds, Leeds, LS2 9JT, UK

[3]Remote Sensing Group, STFC Rutherford Appleton Laboratory, Chilton, Oxfordshire, OX11 0QX, UK

[4]National Centre for Earth Observation, STFC Rutherford Appleton Laboratory, Chilton, Oxfordshire, OX11 0QX, UK

[5]National Centre for Atmospheric Science, University of Leeds, Leeds, LS2 9PH, UK

Submitted to *Atmospheric Chemistry and Physics*

Corresponding author: Richard J. Pope (R.J.Pope@leeds.ac.uk)

**Key Points:**

- The European satellite record shows large lower tropospheric spring-summer ozone reductions in 2020–2022 (8.4-14.6%).

- Scaling precursor emissions based on activity data yields large model ozone reductions in the spring-summer of 2020 and 2021.

- In 2020, meteorology contributed ~1/3 of the modelled reduction (low stratosphere-troposphere flux), with ~2/3 from emission reductions.

**Abstract**

Activity restrictions during the COVID-19 pandemic caused large-scale reductions in ozone ($O_3$) precursor emissions, which in turn substantially reduced the abundance of tropospheric $O_3$ in the Northern Hemisphere. Satellite records of lower tropospheric column $O_3$ (0 – 6 km) from the Rutherford Appleton Laboratory (RAL) highlight these large reductions in $O_3$ during the COVID-19 period (2020), which persisted into 2021 and 2022. The European domain average $O_3$ reduction ranged between 2.0 and 3.0 Dobson units (DU) (11.0-14.6%). These satellite results were supported by the TOMCAT chemistry transport model (CTM) through several model sensitivity experiments to account for changes in emissions and impact of the meteorological conditions in 2020. Here, the business-as-usual (BAU) emissions were scaled by activity data (i.e. anonymised mobility data from big tech companies) to account for the reduction in $O_3$ precursor emissions. The model simulated large $O_3$ reductions (2.0-3.0 DU), similar to the satellite records, where approximately 66% and 34% of the $O_3$ loss can be explained by emissions changes and meteorological conditions, respectively. Our results also show that the reduced flux of stratospheric $O_3$ into the troposphere accounted for a substantial component of the meteorological signal in the overall lower tropospheric $O_3$ levels during the COVID-19 period.

**Plain Language Summary**

Lockdowns and other measures implemented to limit the spread of COVID-19 reduced human activity, leading to a reduction in emissions from humans, including precursors for tropospheric ozone ($O_3$), a pollutant and greenhouse gas. Studies have shown a reduction in tropospheric $O_3$ across the northern hemisphere in the spring-summer of 2020, which coincided with this emission reduction. We provide further evidence of the tropospheric $O_3$ reduction in 2020, specifically for Europe, using two records derived from satellite instruments. The two records show large reductions in European tropospheric $O_3$ in the spring-summer of 2020, peaking at ~ 10 – 20 % in May. We use a chemical transport model to distinguish between the impacts of emissions and meteorology in 2020–2021. In both years, emission reductions had greater influence on the $O_3$ reduction (~2/3), highlighting the importance of emissions in decreasing $O_3$. However, emissions reductions alone were not responsible for the large $O_3$ reduction, as there was considerable influence from meteorology (~1/3), mostly from variation in the flux of $O_3$ from the stratosphere.

**1 Introduction**

Tropospheric $O_3$ is an important secondary atmospheric pollutant and short-lived climate forcer, formed in the presence of the precursor gases, nitrogen oxides ($NO_x$, referring to nitrogen dioxide ($NO_2$) and nitric oxide (NO)) and volatile organic compounds (VOCs), and sunlight (P. S. Monks et al., 2015). Tropospheric $O_3$ is a persistent health problem in Europe, with 24,000 premature deaths attributed to acute $O_3$ exposure in 2020 (European Environment Agency, 2022). $O_3$ is also the 3rd most important greenhouse gas, with an estimated effective radiative forcing of 0.47 W m$^{-2}$ (0.24–0.71 W m$^{-2}$) between 1750–2019, dominated by changes in tropospheric $O_3$ (IPCC, 2021; Skeie et al., 2020).

Due to a global pandemic caused by COVID-19 (disease from SARS-CoV-2, severe acute respiratory syndrome
coronavirus-2), many countries worldwide implemented a 'lockdown' of daily life activities to prevent the spread of
the disease (Forster et al., 2020; WHO, 2020; Zhou et al., 2020). This resulted in a widespread reduction in
anthropogenic surface emissions, including $O_3$ precursor gases. Based on activity data, Forster et al. (2020)
estimated a global reduction of ~ 30% for $NO_x$, 25% for carbon monoxide (CO) and 20% for VOCs in April 2020
and Guevara et al. (2021) estimated reductions of ~ 33% for $NO_x$ and 8% for VOCs in March/April 2020. Here, the
changes in activity data reported by Forster et al., (2020) are based on changes in anonymised mobility data (e.g.
from phone GPS information) provided by Apple and Google (see Forster et al., (2020) and references within).
Typically, they found these mobility datasets used in their study to be within 20% of each other and had a
correlation of 0.8 or higher. Furthermore, Guevara et al. (2021) found that countries with the severest lockdowns had
even higher average reductions (~ 50% for $NO_x$, 14% for VOCs).

Reductions in tropospheric $O_3$ in the spring-summer across the northern hemisphere (NH) free troposphere (FT) was
initially described by Steinbrecht et al. (2021). The timing of this reduction coincides with the introduction of
lockdowns across Europe, beginning in the spring-summer of 2020 and continuing into 2021. Steinbrecht et al.
(2021) found that in 2020, measurements of the NH FT (mostly from ozonesondes) from April–August showed ~7%
lower $O_3$ values, compared to its climatology of 2000–2020. Such a widespread reduction occurring at so many
stations had not occurred previously in this time period. Another notable event during winter-spring of 2019/2020
was the very large stratospheric Arctic $O_3$ depletion caused by a very cold, strong and long-lasting polar vortex (W.
Feng et al., 2021; Weber et al., 2021; Wohltmann et al., 2020). Steinbrecht et al. (2021) suggested that this low
stratospheric $O_3$ event contributed to less than 25% of this $O_3$ negative anomaly, attributing most of the $O_3$ reduction
to emission reductions. Further studies have confirmed low FT $O_3$ across Europe and the NH using aircraft and
ozonesonde measurements (e.g. Chang et al. (2022); Clark et al. (2021); Putero et al. (2023). In contrast, Parrish et
al. (2022) suggested that low 2020 tropospheric $O_3$ could be largely due to a negative trend in baseline tropospheric
$O_3$ since around the mid-2010s, based on Western European surface sites.

From a satellite perspective, Ziemke et al. (2022) found low NH spring-summer FT $O_3$ from instruments aboard
NASA satellites, using a merged instrument record. The tropospheric column $O_3$ reduction of ~ 7–8% (3 DU)
(compared to 2016–2019), was comparatively uniform between 20°N - 60°N and repeated in the next year, 2021.
They found a reduction of NH satellite-derived $NO_2$ (~ 10–20%) in the spring-summer of 2020 and 2021, attributing
this as the likely cause of the $O_3$ reduction. Cuesta et al. (2022) found that satellite-derived lowermost tropospheric
$O_3$ (< 3 km altitude) in the spring ($1^{st}$–$15^{th}$ April) of 2020 was enhanced across central Europe and northern Italy
(typically VOC-limited regions) compared to the previous year (2019) and reduced elsewhere in Europe (typically
$NO_x$-limited regions). An enhancement of $O_3$ across central Europe in the spring-summer of 2020 was also found at
surface monitoring sites (e.g. Ordóñez et al. (2020); Grange et al. (2021)). Apart from Ziemke et al. (2022), there are
few studies of 2021 and onwards. One example is from Pey & Cerro (2022), finding reduced background $O_3$ values
over SW Europe (~15% at most sites) in March-April 2020, which was also seen in 2021 but to a lesser extent.
Similar results were found in the study by Dunn et al., (2024).

Modelling studies have investigated the impact of emission reduction on FT $O_3$, using different methods to estimate
the size of these emission reductions, which are still uncertain. Bouarar et al. (2021) modelled primary pollutant
emission reductions, based on emission reductions from activity data by Doumbia et al. (2021), finding zonally
averaged NH FT $O_3$ to be reduced by 5–15% (2001–2019 baseline). One third of this reduction is attributed to
reductions in air traffic, one third is attributed to a reduction in surface emissions and the final third is attributed to
meteorology, including the low 2020 springtime Arctic stratospheric $O_3$. Miyazaki et al. (2021) used data
assimilation, finding a reduction in the global tropospheric $O_3$ burden of ~ 2% in May and June 2020.

Here, we present an update to the European tropospheric $O_3$ record using two satellite products, extending the record
to mid-2023, and present the reductions in the lower FT compared to previous years. Using a 3-D chemical transport
model, TOMCAT (Monks et al., 2017), we explore the impact of scaling the anthropogenic surface emissions (from
activity data changes) on European tropospheric $O_3$ in 2020 and 2021. Lastly, we quantify the relative contribution
of emissions and meteorology to the modelled reduction in tropospheric $O_3$.
**2 Data and Methods**

2.1 Tropospheric Ozone Satellite Datasets

We present satellite-derived $O_3$ from two satellite instruments, the Infrared Atmospheric Sounding Interferometer
(IASI) and the Global Ozone Monitoring Experiment-2 (GOME-2), both aboard EUMETSAT's satellite MetOp-B
(Clerbaux et al., 2009; Munro et al., 2016). The MetOp series of satellites have a sun-synchronous, near polar orbit
with an equator crossing time of 9:30 local solar time (LST). IASI has a swath width of 2200 km, and in the nadir
viewing mode, there are four circular fields of view across-track with a diameter of 12 km, covering a square $50 \times$
$50 \text{ km}^2$ which is scanned across the swath. IASI measures in the infrared (IR) wavelengths (645–2760 cm$^{-1}$) with a
spectral resolution of 0.3 - 0.5 cm$^{-1}$ (Clerbaux et al., 2009). GOME-2 measures in the ultraviolet-visible (UV-Vis)
wavelengths (240–790 nm) with a spectral resolution of 0.26 - 0.51 nm, and has a swath width of 1920 km. The field
of view is scanned across-track yielding 24 ground-pixels of dimension 80 km (across-track) × 40 km (along-track)
(Callies et al., 2000; Munro et al., 2016). For quality assurance, the GOME-2B record was filtered for a geometric
cloud fraction of <0.2 (e.g. Miles et al. (2015)) and the IASI-IMS-extended record was filtered for an effective cloud
fraction of <0.5 (as in Pope et al. (2021)). Here, the RAL Space GOME-2 and IASI Infrared and Microwave
Sounding (IMS) retrieval schemes for lower tropospheric ozone have been independently evaluated against
ozonesonde data in Miles et al., (2015) and Pimlott et al., (2022).

Height-resolved $O_3$ distributions are retrieved by the Rutherford Appleton Laboratory (RAL) using the IMS-
Extended scheme for IASI (detailed in Pope et al. (2021)) and UV-Vis scheme for GOME-2 (detailed in Miles et al.
(2015)). Due to an underlying negative tendency in the GOME-2 record, likely from UV degradation of the

instrument, we have detrended that record, as shown in **Supplement Text S1** and **Figure S1**. To compare the IASI-IMS-Extended data from MetOp-B (2018–2023) to a longer time-period, we combine the record with IASI-IMS-Extended data from MetOp-A (2008–2017). The MetOp-B record was adjusted according to monthly differences with the MetOp-A record in the overlap year of 2018, as described in **Supplement Text S2** and **Figures S2 and S3**. Here we use lower tropospheric sub-columns of the surface–450 hPa (~6 km altitude) derived from the retrieved profiles, with a focus on Europe. As such, we use a land mask to extract a terrestrial European signal given the direct link between surface $O_3$, precursors gases and air pollution exposure (see **Supplement Figure S4**).

### 2.2 Model Simulations

We use the TOMCAT 3-D chemical transport model to simulate tropospheric $O_3$ between 2017 and 2021. The model control simulation is for 2017, 2018 and 2019. However, in 2020, the control simulation splits into two scenarios: 1) business-as-usual scenario (BAU) and 2) scaled emission scenario (COVID). For the BAU scenario, the control modelled emissions inventory is used but for the COVID scenario, we apply emission reduction factors (Forster et al., 2020) to model surface and aircraft emissions to account for changes in activity due to the pandemic in 2020 and 2021. However, COVID scaling for emissions are not available beyond 2021, so the model simulations are restricted to 2017-2021. TOMCAT is an off-line model driven by 6-hourly ERA-5 meteorological reanalyses (e.g. temperature, relative humidity, winds; Hersbach et al., 2020), which are provided by the European Centre for Mid-Range Weather Forecasting (ECMWF). The ERA-5 meteorological reanalyses are provided on 137 vertical levels (surface to 1 hPa), which are interpolated onto the TOMCAT vertical grid (31 levels -see Monks et al., (2017) Figure 1). It has a horizontal a resolution of $2.8° \times 2.8°$ and 31 vertical levels between the surface and 10 hPa, coupled with the Global Model of Aerosol Processes (GLOMAP) (Chipperfield, 2006; Mann et al., 2010; Spracklen et al., 2005). The chemistry scheme includes approximately 80 advected tracers and over 200 chemical reactions (S. A. Monks et al., 2017).

Surface emission fields are described in detail in **Supplement Text S4** and **Table S1**. The anthropogenic emissions are from the Coupled Model Intercomparison Project Phase 6 (CMIP6) (L. Feng et al., 2020), whereby after 2014 emissions are based on Shared Socioeconomic Pathways (SSPs) (Gidden et al., 2019; Riahi et al., 2017). In this study, we have used the middle-of-the-road scenario, SSP2-4.5, for the TOMCAT control run between 2017 and 2019, before diverging into the BAU and COVID simulations. For the BAU simulation, the CMIP6 SSP2-4.5 emissions are used but for the COVID simulation, scaling factors for emission reductions from national lockdowns come from Forster et al. (2020) and were applied to the BAU emissions. Forster et al. (2020) used national mobility/activity data to estimate reductions in air pollutant emissions (i.e. $NO_x$, CO, VOCs, black carbon (BC) and organic carbon (OC)). **Figure 1(a)** highlights the impacts of the scale factors, with substantial decreases evident in European emissions for $NO_x$, CO and VOCs. **Figure 1(b)** shows that the peak reductions were in April 2020, once most European lockdowns were in effect, with monthly reductions of 0.44 Tg (33%), 0.75 Tg (34%) and 0.06 Tg (29%) of $NO_x$ (as $NO_2$), CO and NMVOCs (as carbon (C)), respectively. For 2020, a secondary winter emissions reduction occurs at ~ 15–20% as further European lockdowns were imposed to reduce the spread of COVID-19. For

2021, the scaling factors from Forster et al. (2020) suggest that emissions were approximately 10–13% lower than
expected but remained consistent throughout the year, suggesting a potential 'new normal' of lower precursor
emissions. A tracer for stratosphere-troposphere exchange (STE) in the model ($O_{3S}$) is used to understand the impact
of $O_3$ transport from the stratosphere. In the stratosphere, it is set equal to the model-calculated $O_3$. The only
tropospheric source of the tracer is transport from the stratosphere while its sinks are via photolysis, surface
deposition and reactions with $HO_2$, OH and $H_2O$ through $O(^1D)$ produced from $O_{3S}$ (S. A. Monks et al., 2017).

Overall, TOMCAT is a robust and well evaluated CTM having been used in multiple studies of tropospheric $O_3$ and
compared with many types of observation (e.g. Richards et al., (2013), Pope et al., (2020) and Pope et al., (2023,
2024). The simulated tropospheric ozone burden is a common metric to assess the skill of a model to simulate
tropospheric ozone. Here, we derive a tropospheric $O_3$ burden of 322 Tg (BAU 2020 simulation), which is
consistent with that of Monks et al., (2017) who reported an equivalent of 331 Tg. Both estimates sit within the
reported range of 337±23 Tg from the Atmospheric Chemistry and Climate Model Intercomparison Project
(ACCMIP, Young et al., 2013) further demonstrating TOMCAT to be a suitable modelling framework. Highly
relevant for this work, Pope et al., (2023) included a detailed comparison of lower tropospheric ozone between
TOMCAT and GOME-2/IASI, where thorough consideration of the satellite averaging kernels (i.e. function of
satellite vertical sensitivity when retrieving sub-column profiles of $O_3$) was taken in conjunction with the model,
generally displaying good agreement with the between them. Therefore, we are confident in using TOMCAT to
directly investigate the impact of COVID-19 on lower tropospheric ozone over Europe.

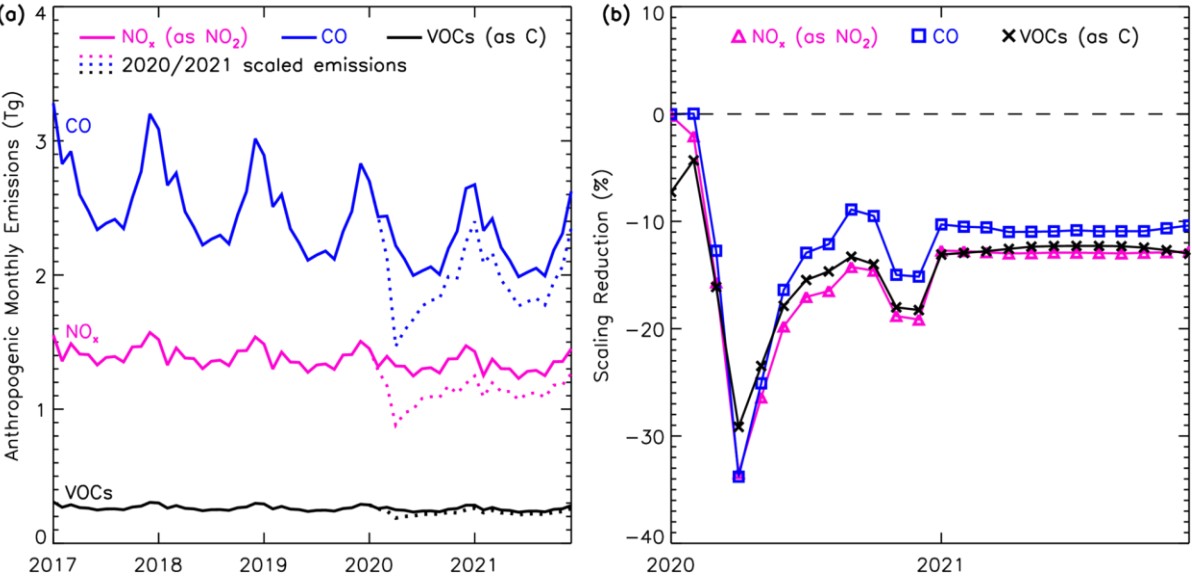

**Figure 1.** European aggregated anthropogenic monthly emissions of $NO_x$ (as $NO_2$), CO and NMVOCs (as C) used
in the TOMCAT simulations between 2017 and 2021. (a) BAU emissions (solid) and COVID emissions in 2020 and
2021 (dotted) (Tg). (b) Percentage reduction in 2020 and 2021 for $NO_x$, CO and VOCs in the COVID emissions,
relative to the BAU emissions.

**3 Results and Discussion**

3.1 European Tropospheric Ozone Satellite Record (2008–2023)

We present two satellite-derived lower tropospheric sub-column $O_3$ records for continental Europe from 2008–2023 (**Figure 2**). During the overlapping years of 2015–2019, the records show an average difference of 2.5 DU, but the variability is well correlated (Pearson's correlation coefficient ~ 0.80). Satellite record inconsistencies are likely due to differences between IR and UV-Vis instruments, the related retrieval schemes and their vertical sensitivities, despite the instruments being aboard the same platform and having the same overpass time. Compared to a monthly baseline of 2015–2019 for GOME-2B and 2008–2019 for IASI-IMS-Extended, the monthly anomalies (**Figure 2(b)**) show good agreement through this overlap period, with the most notable disagreements in winter/spring of 2015 and spring-summer 2016. Both records show large negative anomalies in spring-summer 2020. GOME-2B shows peak negative anomalies of 2.4 DU (18.3%) and 3.0 DU (21.4%) in April and May 2020, respectively, and IASI-IMS-Extended shows slightly smaller negative anomalies of 1.7 DU (9.4%) and 2.2 DU (11.0%) in April and May, respectively. For the records shown in **Figure 2(b)**, two standard deviations ($2\sigma$) across the entire monthly record is 2.1 DU for GOME-2B and 1.8 DU for IASI-IMS-Extended. Thus, ~ 95% of the data ranges between the average $\pm 2\sigma$ for the respective records. In both cases, April and May 2020 negative anomalies either match or surpass this range signifying relatively substantial anomalies for these months, highlighting their unusual nature. The reductions continue into the summer of 2020, with both records showing large negative anomalies in July and August: 1.7 DU (9.2%) and 1.4 DU (7.2%) for GOME-2B; 1.8 DU (8.3%) and 1.3 DU (6.3%) for IASI-IMS-Extended.

Tropospheric $O_3$ reductions continue into the spring and summer period of 2021, with the IASI-IMS-Extended record showing negative anomalies in most months of 2021, however, these anomalies are slightly smaller than in 2020. The largest negative anomalies are in April, May and June, at 1.0 DU (5.3%), 1.7 DU (8.4%) and 1.1 DU (5.2%), respectively, with only the reduction in May being close to the average $\pm 2\sigma$ threshold. This recurrence in 2021 of a tropospheric $O_3$ reduction of similar magnitude to 2020 is consistent with the combined NASA satellite product tropospheric column $O_3$ record for the 20-60N latitude band reported by Ziemke et al. (2022), which is presented from January–August.

It is worth noting that there is approximately a 1-month lag between the IASI and GOME-2 time-series in Figure 2 which is likely due to the European domain (see Figure S4 of the Supplement) extending to high northern latitudes (approximately 70°N) where sampling of the GOME-2 UV sounder, but not IASI, is restricted in winter months by absence of sunlight. While this could slightly influence the domain average annual cycle comparison it does not affect the interannual variability subject of this study.

In 2022, the IASI-IMS-Extended record shows even larger negative anomalies in April and May than 2020/2021, of 2.6 DU (15.0%) and 2.8 DU (14.6%), respectively, which are well beyond the average $\pm 2\sigma$ threshold. The negative

anomalies continue in June and July, with 1.3 DU (6.5%) and 1.3 DU (6.1%). In 2023, the negative
spring-summer are smaller compared to 2020–2022, apart from in May, where the negative anomaly is 1.3 DU
(6.1%). Broadly, the years of 2020–2023 all show monthly anomalies which are more consistently negative than the
previous 12 years. The question of the persistence of low European $O_3$ values will become evident in future years
through extension of these MetOp records.

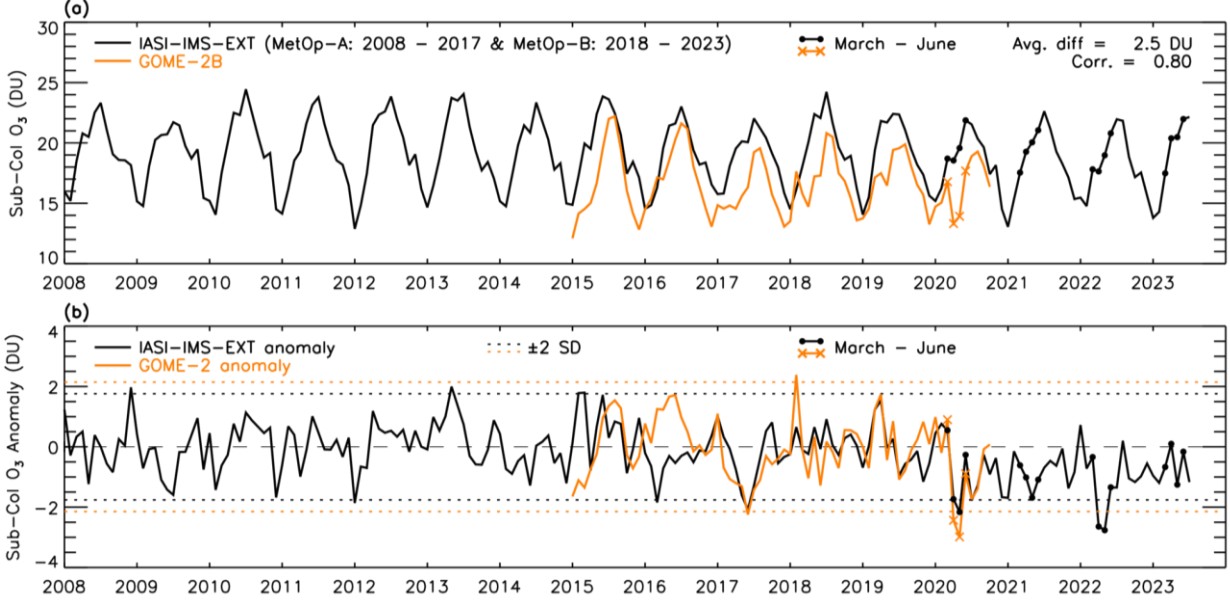


**Figure 2.** European satellite-derived $O_3$ from January 2008–July 2023. (a) Monthly average sub-column (surface–
450 hPa) $O_3$ record (DU) from IASI (IASI-IMS-extended, January 2008–July 2023) and GOME-2B (January 2015–
October 2020). (b) Monthly mean anomalies for the two records (2015–2019 baseline for GOME-2B, 2008–2019
for IASI-IMS-Extended) (DU). Dotted lines indicate ± 2σ from the average of the record. Filled circles (IASI-IMS-
Extended) and crosses (GOME-2B) are shown for the months of March–June in 2020–2023, to highlight the
relevant spring/summer periods. Average difference and correlation are based on January 2015–December 2019.

3.2 TOMCAT Model Experiments (2017–2021)

In 2020, scaling the emissions according to the mobility data estimates in Forster et al. (2020) (TOMCAT COVID
scenario) caused a monthly reduction in tropospheric $O_3$ from March to December (**Figure 3(a)**). During January
and February, the COVID and BAU scenarios are very similar, however, from March onwards the COVID scenario
shows a negative difference compared to the BAU scenario, which peaks at 2.0 DU (8.3%) lower in May. This
negative difference then reduces through the year to December (0.7 DU, 4.1%). **Figure 4a** shows the spatial impact
of COVID-19 on lower troposphere ozone simulated by TOMCAT. The March-May 2020 average is typically 1.0-
2.0 DU lower across the whole European domain. In 2021, the COVID scenario in Figure 3(a) shows consistent
reductions in all months of the year, starting at 0.6 DU (3.4%) in January, peaking at 1.0 DU (4.3%) in May, and
reducing towards the end of the year, ending with 0.6 DU (3.2%) in December. The temporal pattern of the
reduction is similar to that in surface emissions (**Figure 1**), although with considerably smaller percentage decreases
(peak of ~30% for surface emissions and ~8% for the resulting $O_3$ sub-column). This highlights the large emission
reductions required for a sizeable reduction in European lower tropospheric $O_3$. To identify the impact of
meteorology in 2020, the scaled emissions in 2020 were used in three separate simulations with the meteorology of
2017, 2018 and 2019, with an average of these three scaled emission simulations shown in **Figure 3(b)**. The 2020
COVID scenario record is broadly lower than the 2017/2018/2019 averaged scaled emission scenario, despite using
the same surface emissions, which indicates that the meteorology of 2020 had a large impact on the tropospheric $O_3$
reduction. Here, we use the term "meteorology" to represent meteorological variables such as temperature, pressure
and humidity, but also the long-range transport (i.e. advection/convection) of air masses, which influence
tropospheric chemistry. This is supported by **Figure 4b** which shows that across most of Europe, 2020
meteorological conditions where more conducive to lower tropospheric ozone loss (i.e. differences of -3.0 and -1.0
DU) than previous years. However, the domain average shown for March-May 2020 in **Figure 3b** is buffered by the
positive differences (up to 1.0-1.5 DU) above 60$^\bullet$*N*. The impact of meteorology in 2020 is greatest in the spring-
summer (**Figure 3b**), as the differences between these two timeseries is largest from February–July, peaking at a 1.1
DU difference in May. This demonstrates the importance of meteorology to the resulting $O_3$ in the spring-summer of
2020. The records are much more consistent from August to the end of the year, with absolute differences below 0.6
DU, indicating a reduced impact from meteorology in the second half of the year.
In comparison with the previous 3 years (2017–2019), the BAU scenario in 2020 and 2021 has lower peak spring-
summer values of $O_3$, especially compared to the high $O_3$ values in 2019 (**Figure 3(a)**). The spring-summer of 2020
shows negative anomalies in the BAU scenario of up to 1.4 DU (-5.8%) (**Figure 3(c)**). April, May and July show the
largest reductions, which are around the value of the average $\pm 2\sigma$ threshold ($\pm$ 1.3 DU, 6.2%). The spring-summer
BAU scenario reductions are repeated in 2021 from January–June, peaking at 1.2 DU (4.9%) in May. Any variation
in the BAU scenario is due to meteorology and also variation in the BAU surface emissions used. As shown in
**Figure 1**, the BAU emissions only vary by a small amount from year to year, e.g. the average total annual
anthropogenic emission difference between consecutive years across the simulation time period is 0.33 Tg (2.0%)
for $NO_x$, 1.3 Tg (4.4%) for CO and 0.06 Tg (1.8%) for VOCs. With consistent BAU emissions, meteorology is the
dominant control in the BAU scenario and had a large impact on the simulated tropospheric $O_3$ in the spring and
summer of 2020.

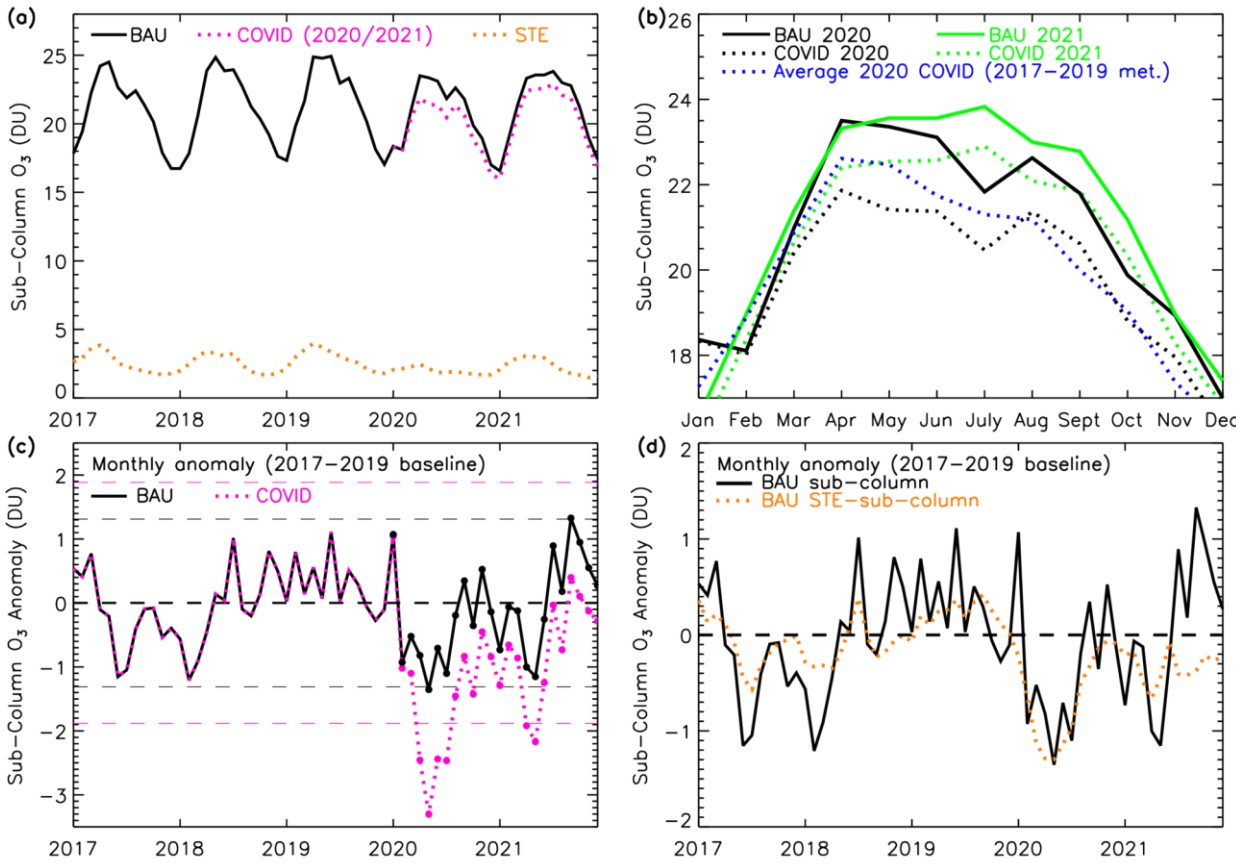

**Figure 3**. TOMCAT European lower tropospheric sub-column $O_3$ (surface–450 hPa) between 2017 and 2021 (DU). (a) Monthly sub-column $O_3$ averages for the BAU sub-column (solid, black) and STE-contribution sub-column (dotted, orange). The COVID scenario is shown in 2020 and 2021 (pink dotted). (b) BAU (solid, black) and COVID scenario (dotted) records for 2020 (black), 2021 (green), with the 2017/2018/2019 averaged COVID scenario (2020 scaled emissions, dark blue, dotted). (c) BAU (solid, black) and COVID (pink, dotted) $O_3$ anomalies (baseline of 2017–2019). Horizonal dashed lines indicate ± 2σ from the average of the record. (d) As panel (c) with the inclusion of monthly $O_3$ anomalies of the stratosphere-troposphere exchange (STE)-contribution sub-column (orange, dotted).

The COVID scenario shows large negative anomalies in 2020, peaking at 3.3 DU (15.4%) in May 2020 (**Figure 3c**), which is much more than the average ± 2σ threshold (± 1.9 DU, 9.0%). Comparing the BAU and COVID scenarios suggests that ~1 DU of the negative anomaly is due to meteorology (and small variations in BAU emissions) and the remaining contribution (~ 1–2 DU in spring-summer) of the negative anomaly is due to the scaled emissions for 2020. The contribution of $O_3$ from STE to the troposphere in the model sub-column is calculated by TOMCAT as a tracer which represents stratospheric $O_3$ that has entered the troposphere and is controlled by tropospheric sink processes. We calculate a sub-column based on this contribution (STE-sub-column), shown in **Figure 3(a)**, varying between 1.5–4.0 DU from 2017–2021. We find a large negative anomaly in model stratosphere-troposphere $O_3$ exchange (STE) in the spring-summer of 2020 (**Figure 3(d)**), of 1.3 DU in both April and May (52.5% and 60.5%, respectively). The STE-sub-column absolute negative anomaly is a similar value or larger than the lower tropospheric sub-column anomaly from March - August in 2020, suggesting that during this period, low STE

contribution was a substantial factor in the BAU scenario lower tropospheric sub-column $O_3$ reduction. In the
months where the STE-sub-column absolute anomaly is larger than the BAU anomaly, the other controlling factors
in the BAU simulation $O_3$ are likely around neutral or even slightly positive. The stratospheric $O_3$ used in the model
simulation is a climatology, therefore, any variation on the STE contribution is from variation in the STE flux. In
2021, the negative anomaly in STE-sub-column is smaller than for 2020, reaching a peak value of 0.7 DU (21.5%)
in April (**Figure 3(d)**). The STE-sub-column negative anomaly is also not larger than for the lower tropospheric
sub-column in 2021, suggesting that the STE reduction had a smaller impact on the negative lower tropospheric sub-
column anomalies seen in 2021, in comparison with 2020.

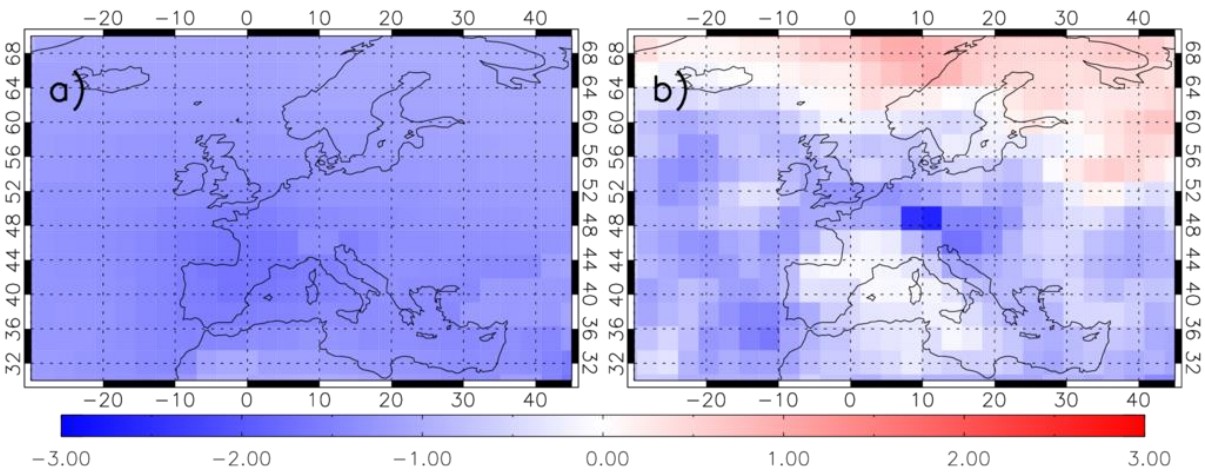

**Figure 4**: TOMCAT lower tropospheric ozone (DU) differences (March-May 2020 average) between a) the
TOMCAT COVID and TOMCAT BAU simulations and b) the TOMCAT COVID simulation with 2017-2019
average meteorology (TOMCAT run for 2017, 2018 and 2019 with 2020 COVID emissions and the three
simulations averaged together) and the TOMCAT BAU simulation.
To further quantify the relative contributions, the difference between the anomalies for the BAU and COVID
scenario as a relative percentage of the COVID scenario for 2020 (i.e. $100 \times$ (BAU - COVID)/COVID) is shown in
**Figure 5(a)**. We performed this quantification for spring-summer months showing a negative anomaly in both
scenarios (March–August 2020 and March–June 2021). These values represent the percentage contributions of the
emission reductions (due to COVID-19) and meteorological conditions to the determined reduction in the lower
tropospheric column zone. The contribution of emissions to the COVID scenario in spring-summer 2020 is 53%
(March), 67% (April), 59% (May), 71% (June), 55% (July) and 87% (August), with an average of 65% across these
months. Therefore, scaling the emissions is the dominant influence during this period. In 2021, the COVID scenario
also shows large negative anomalies, peaking at 2.2 DU (9.6%) in May. Scaling the emissions contributed towards
86%, 48%, 47% and 80% for March–June, respectively, of the scaled negative anomaly (average of 65%), with the
rest due to meteorology (and BAU emissions).



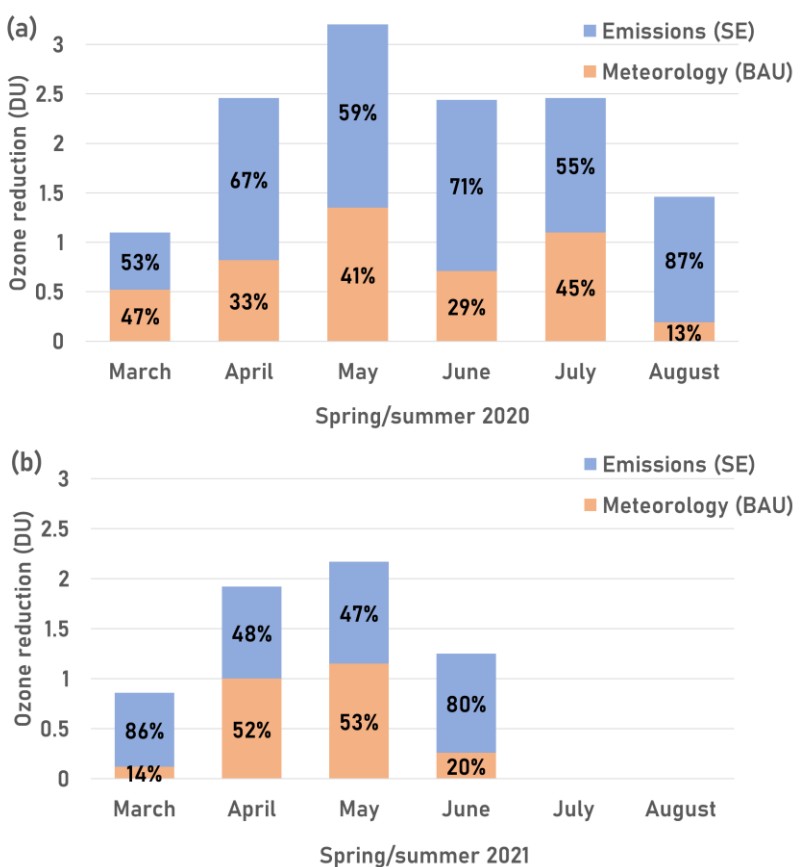

**Figure 5.** Contribution of scaled emissions and meteorology/BAU emissions to the TOMCAT lower tropospheric
sub-column $O_3$ reduction from March–August in (a) 2020 and (b) 2021. The total reduction (DU) is the COVID
scenario negative anomaly, with the relative contribution of meteorology/BAU emissions shown in orange and the
contribution of scaled emissions shown in blue. The percentage relative contribution is labelled onto each bar
section.
**4 Conclusions**
Our study represents one of the first extended investigations of the COVID-19 pandemic impacts on European lower
tropospheric $O_3$ (surface-450 hPa) using satellite observations and modelling. The records from the Global Ozone
Monitoring Experiment (GOME – 2) and the Infrared Atmospheric Sounding Interferometer (IASI) on MetOp-B
show substantial decreases in European average spring-summer time lower tropospheric ozone of typically 2.0-3.0
DU (or 11.0-14.6%). While not the key focus of this paper, the 2022 decline in $O_3$ is interestingly the largest
between 2020 and 2023. Therefore, this would suggest other factors not investigated in this study are driving a more
substantial $O_3$ decrease and that the reported COVID-19 response is within the more extreme variability of European
ozone.
To investigate the drivers of the $O_3$ decreases over Europe during the COVID-19 period (2020-2021), activity
scaling factors (i.e. based on anonymised mobility data from big tech firms) were used to perturb the model's
business-as-usual (BAU) emissions for 2020 and 2021 to quantify the COVID-19 impact on $O_3$. Here, the
TOMCAT simulations of lower tropospheric $O_3$ were reduced by 2.0-3.0 DU (comparable to the $O_3$ reductions
reported by the satellite records) in the COVID-19 simulation compared to the BAU baseline. Further model
sensitivity experiments were able to diagnose the contribution of 2020 emissions changes (approximately 66%) and
2020 meteorological conditions (approximately 34%) to the overall TOMCAT simulated $O_3$ reduction in Europe.
Therefore, the COVID-19 reduced in $O_3$ precursor emissions were substantial in reducing 2020 European $O_3$, but it
was amplified by meteorological conditions that year. Investigation of the TOMCAT stratospheric $O_3$ tagged tracer
(i.e. a representation of the flux of stratospheric rich $O_3$ air into the troposphere) suggested a substantial drop in its
contribution to lower tropospheric $O_3$ (in the order of 1.0 DU), which was comparable to the meteorological signal.
Thus, a likely cause of the amplified European $O_3$ reduction in the COVID-19 period.
Therefore, our study has successfully quantified the impact of COVID-19 on European lower tropospheric ozone
and identified a useful methodology to isolate the impact of emission changes, but also importantly meteorological
variability, on observed changes in tropospheric composition. Future work would focus on the large reduction in
European $O_3$ in 2022 (which is beyond the scope of this study), produce a harmonised IASI $O_3$ record from the three
MetOp satellites and a reprocessing of the RAL Space GOME-2 record to more accurately account for UV-
degradation in the instrument record.

**Acknowledgements**
This work was funded by the UK Natural Environment Research Council (NERC) by providing funding for the
National Centre for Earth Observation (NCEO, award reference NE/R016518/1) and the NERC Panorama Doctoral
Training Programme (DTP, award reference 580 NE/S007458/1). The TOMCAT runs were undertaken on ARC3,
part of the High-Performance Computing facilities at the University of Leeds, UK.
**Data Availability**
The IASI-IMS and GOME-2 data is available via the NERC Centre for Environmental Data Analysis (CEDA)
Jasmin platform subject to data requests. However, the IASI-IMS data and TOMCAT simulations used in this study
are available on Zenodo at https://zenodo.org/records/10424302 (Pimlott et al., 2024).
**Author Contributions**
MAP and RJP conceptualised, planned and undertook the research study. BJK, RS, BGL and LJV provided the data
and advice on using the products. MAP performed the TOMCAT model simulations with support from MPC and
WF. MAP prepared the manuscript with contributions from all co-authors.
**Conflicts of Interest**
The authors declare no conflicts of interest.

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
