# Peer review of "Large Reductions in Satellite-Derived and Modelled European Lower Tropospheric"

_EGUsphere, 2024_

## Referee Comment (RC1)

The manuscript entitled "*Large Reductions in Satellite-Derived and Modelled European Lower Tropospheric Ozone During and After the COVID-19 Pandemic (2020–2022)*" presents satellite records of European tropospheric O3 from pre-COVID period to mid-2023 (post-COVID). In addition to providing evidence of O3 reduction during the COVID-19 pandemic – consistent with findings from previous studies – the authors report an even larger reduction in O3 in 2022, after the pandemic. This finding could hold scientific importance if further investigation into its underlying causes is conducted. Alongside the analysis of satellite products, the authors utilized a chemical transport model to separate the contributions of emissions and meteorology to $O_3$ reduction from 2020 to 2021. Their results suggest that meteorological factors, particularly stratosphere-troposphere exchange, play a crucial role in $O_3$ reduction. However, I find that the conclusions drawn in this section lack robustness due to the following reasons: 1. There is no $O_3$ budget analysis, which is necessary for a comprehensive evaluation of contributing factors. 2. A more thorough discussion of uncertainties related to the scaling factor, model resolution, and other relevant aspects is needed to enhance the scientific rigor of the study. Overall, the manuscript may be suitable for publication after major revision.

**General comments**:

1. Depth of scientific analysis: The current manuscript primarily presents data analysis and modeling results but does not sufficiently explore the underlying scientific implications. The discussion/conclusion sections are brief and lack a deeper interpretation of the findings. I recommend expanding these sections to provide more context, including possible explanations, comparisons with previous studies, and insights into the broader scientific relevance of the results. For example, the finding that the largest O3 reduction occurred in 2022, after the COVID-19 period, could hold scientific significance if potential explanations for this trend were explored in greater depth.

2. Uncertainty analysis and discussion: The analysis presented in this study generally lacks an assessment of uncertainty, which is crucial for interpreting and validating the results. Including uncertainty analysis in Section 3 and discussing limitations in the conclusion would enhance the robustness of the findings. Currently, the conclusion drawn from the modeling analysis is limited due to absence of uncertainty analysis. I recommend a more thorough discussion on uncertainties related to the scaling factor used, model resolution, etc. These additions would significantly improve the scientific rigor of the study.

**Specific comments**:

1. Line 136: If the authors have done a literature review but still lack of an emission scaling factor for 2022, running a BAU case with 2022 meteorological data could provide

valuable modeling insights into the causes behind the observed large O3 reduction in 2022. This would significantly enhance the scientific impact of this study.

2. Line 156: The implementation of the STE tracer is somewhat difficult to understand. It seems unusual that the only chemical sinks, aside from photolysis, are reactions with HO2/OH and H2O through O1D produced from *the tracer itself.* Could you provide more explanation on how this tracer could represents the O3 budget or its contribution to tropospheric O3? Alternatively, citing previous studies that use a similar tracer implementation could help, too.

3. Fig2(a): In examining the sub-col O3 time series from the two satellite products, I noticed that, in addition to a generally lower values across the observation period – which the authors suggest might be due to instrument biases – GOME-2B appears to have a one-month phase compared to the IASI data. For instance, GOME-2B consistently shows a lower trough at the end of 2015, 2016, 2017, 2018, and 2019, whereas IASI shows this trough at the beginning of each respective year. Could you offer some potential reasons for this discrepancy?

4. Line 223 (Figure 3(b)): Given that the represent an average of three simulations, I suggest adding error bars to indicate the standard deviation in the plot alongside the averaged values. This would help illustrate the significance of the differences more effectively.

5. Line 251: "*These values represent the contribution of emission reduction.. and the corresponding contribution of meteorology*" – do you mean that the ratio represents the contribution of emission reduction, while (*1- ratio)* represents the contribution of meteorology? Please consider rephrasing this sentence to improve clarity and readability.

6. Line 260: As mentioned previously, the definition of STE is not very clear. Could you elaborate further on why it represents the contribution of O3 from STE? Specifically, can the STE tracer defined in this study be transferred as contribution of O3 from STE? For instance, Griffiths et al., 2020 validate their derived STE using O3 budget: STE = LO3+DO3-PO3. Could you adopt a similar method to demonstrate that the STE used in this paper aligns with this O3 budget equation? Additionally, I recommend including a detailed O3 budget analysis to provide a more robust and concrete evaluation of contributing factors.

7. Line 267: Concluding that other factors are likely neutral or even positive based solely on the STE tracer anomaly being larger than the O3 anomaly may not be very robust. An O3 budget analysis could provide a clearer understanding of the contributions from various

factors, such as chemical production, chemical losses and deposition losses, all of which are influenced by meteorological parameters and affect O3 concentrations.

**References**:

Griffiths, P. T., Keeble, J., Shin, Y. M., Abraham, N. L., Archibald, A. T., & Pyle, J. A. (2020). On the changing role of the stratosphere on the tropospheric ozone budget: 1979–2010. Geophysical Research Letters, 47, e2019GL086901. https://doi.org/10.1029/2019GL086901

---

## Community Comment (CC1)

November 4, 2024

Comments by Owen R. Cooper (TOAR Scientific Coordinator of the Community Special Issue) on:

**Large Reductions in Satellite-Derived and Modelled European Lower Tropospheric Ozone During and After the COVID-19 Pandemic (2020–2022)**

Matilda A. Pimlott, Richard J. Pope, Brian J. Kerridge, Richard Siddans, Barry G. Latter, Lucy J. Ventress, Wuhu Feng, and Martyn P. Chipperfield

EGUsphere [preprint], https://doi.org/10.5194/egusphere-2024-2736
Discussion started Sep. 20, 2024
Discussion closes Nov. 5, 2024

This review is by Owen Cooper, TOAR Scientific Coordinator of the TOAR-II Community Special Issue. I, or a member of the TOAR-II Steering Committee, will post comments on all papers submitted to the TOAR-II Community Special Issue, which is an inter-journal special issue accommodating submissions to six Copernicus journals:  ACP (lead journal), AMT, GMD, ESSD, ASCMO and BG. The primary purpose of these reviews is to identify any discrepancies across the TOAR-II submissions, and to allow the author teams time to address the discrepancies.  Additional comments may be included with the reviews. While O. Cooper and members of the TOAR Steering Committee may post open comments on papers submitted to the TOAR-II Community Special Issue, they are not involved with the decision to accept or reject a paper for publication, which is entirely handled by the journal's editorial team.

**Comments regarding TOAR-II guidelines:**

TOAR-II has produced two guidance documents to help authors develop their manuscripts so that results can be consistently compared across the wide range of studies that will be written for the TOAR-II Community Special Issue.  Both guidance documents can be found on the TOAR-II webpage: https://igacproject.org/activities/TOAR/TOAR-II

*The TOAR-II Community Special Issue Guidelines*:   In the spirit of collaboration and to allow TOAR-II findings to be directly comparable across publications, the TOAR-II Steering Committee has issued this set of guidelines regarding style, units, plotting scales, regional and tropospheric column comparisons, and tropopause definitions.

*The TOAR-II Recommendations for Statistical Analyses*:  The aim of this guidance note is to provide recommendations on best statistical practices and to ensure consistent communication of statistical analysis and associated uncertainty across TOAR publications. The scope includes approaches for reporting trends, a discussion of strengths and weaknesses of commonly used techniques, and calibrated language for the communication of uncertainty. Table 3 of the TOAR-II statistical guidelines provides calibrated language for describing trends and uncertainty, similar to the approach of IPCC, which allows trends to be discussed without having to use the problematic expression, "statistically significant".

**General comments:**

A new paper published in the *TOAR-II Community Special Issue* is highly relevant to this study as it shows that the decrease of ozone observed in the free troposphere during 2020 also extended to the surface, as observed at high elevation monitoring sites in North America and Europe (Putero et al., 2023).

An additional relevant study:  Every year the *State of the Climate* reports provide updates on the global distribution and trends of greenhouse gases, including tropospheric ozone. The most recent report (Dunn et al., 2024) reports the latest findings based on NASA's OMI/MLS tropospheric ozone product (see Figure 2.66 on page S95). The data show an increase of the tropospheric ozone burden (60° S – 60° N) from 2004 to 2019, followed by a drop in ozone in 2020 and a levelling off through 2023.

Lines 76-78
The authors cite a paper that claims that free tropospheric ozone is decreasing all across northern mid-latitudes, but no other study has been able to replicate those results. In contrast, plenty of studies have conducted in-depth analysis of ozone observations in the free troposphere, and do not find a decrease of ozone.  IPCC AR6 (Gulev et al., 2021; Szopa et al., 2021) concluded that free tropospheric ozone has increased since the mid-1990s based, in part, on IAGOS observations in the free troposphere (Gaudel et al., 2020). And follow-up studies have shown that ozone increased in the free troposphere from the mid-1990s to 2019 above Europe and western North America, with a decrease of ozone in 2020 due to the COVID-19 pandemic (Chang et al., 2022, 2023).  These in situ observations fit with the decrease of ozone in 2020 and 2021, as observed by several satellite products (Ziemke et al., 2022; Dunn et al., 2024). Another recent study, submitted to the TOAR-II Community Special Issue, uses the ECHAM6-HAMMOZ global atmospheric chemistry model to show that ozone generally increased across the northern hemisphere from 2004 to 2019, in agreement with the OMI/MLS satellite product (see Figure 4 of Fadnavis et al., 2024).  A NASA study reached similar conclusions (Liu et al., 2022).

Minor Comments

line 50
missing the word "of"?

line 293
missing the word "ozone"

**References**

Chang, K.-L., et al. (2022), Impact of the COVID-19 economic downturn on tropospheric ozone trends: an uncertainty weighted data synthesis for quantifying regional anomalies above western North America and Europe, *AGU Advances, 3*, e2021AV000542. https://doi.org/10.1029/2021AV000542

Chang, K.-L., et al. (2023). Diverging ozone trends above western North America: Boundary layer decreases versus free tropospheric increases. Journal of Geophysical Research: Atmospheres, 128, e2022JD038090. https://doi.org/10.1029/2022JD038090

Cooper, et al. 2020. Multi-decadal surface ozone trends at globally distributed remote locations. Elem Sci Anth, 8: 23. DOI: https://doi.org/10.1525/elementa.420

Dunn, R. J. H., J. Blannin, N. Gobron, J. B Miller, and K. M. Willett, Eds., 2024: Global Climate [in "State of the Climate in 2023"]. *Bull. Amer. Meteor. Soc.*, **105** (8), S12–S155, https://doi.org/10.1175/BAMS-D-24-0116.1.

Fadnavis, S., Elshorbany, Y., Ziemke, J., Barret, B., Rap, A., Chandran, P. R. S., Pope, R., Sagar, V., Taraborrelli, D., Le Flochmoen, E., Cuesta, J., Wespes, C., Boersma, F., Glissenaar, I., De Smedt, I., Van Roozendael, M., Petetin, H., and Anglou, I.: Influence of nitrogen oxides and volatile organic compounds emission changes on tropospheric ozone variability, trends and radiative effect, EGUsphere [preprint], https://doi.org/10.5194/egusphere-2024-3050, 2024.

Gaudel, A., et al. (2020), Aircraft observations since the 1990s reveal increases of tropospheric ozone at multiple locations across the Northern Hemisphere. Sci. Adv. 6, eaba8272, DOI: 10.1126/sciadv.aba8272

Gulev, S.K., P.W. Thorne, J. Ahn, F.J. Dentener, C.M. Domingues, S. Gerland, D. Gong, D.S. Kaufman, H.C. Nnamchi, J. Quaas, J.A. Rivera, S. Sathyendranath, S.L. Smith, B. Trewin, K. von Schuckmann, and R.S. Vose, 2021: Changing State of the Climate System. In Climate Change 2021: The Physical Science Basis. Contribution of Working Group I to the Sixth Assessment Report of the Intergovernmental Panel on Climate Change [Masson-Delmotte, V., P. Zhai, A. Pirani, S.L. Connors, C. Péan, S. Berger, N. Caud, Y. Chen, L. Goldfarb, M.I. Gomis, M. Huang, K. Leitzell, E. Lonnoy, J.B.R. Matthews, T.K. Maycock, T. Waterfield, O. Yelekçi, R. Yu, and B. Zhou (eds.)]. Cambridge University Press, Cambridge, United Kingdom and New York, NY, USA, pp. 287–422, doi:10.1017/9781009157896.004

Liu, J., Strode, S. A., Liang, Q., Oman, L. D., Colarco, P. R., Fleming, E. L., et al. (2022). Change in tropospheric ozone in the recent decades and its contribution to global total ozone. Journal of Geophysical Research: Atmospheres, 127, e2022JD037170. https://doi.org/10.1029/2022JD037170

Putero, D., Cristofanelli, P., Chang, K.-L., Dufour, G., Beachley, G., Couret, C., Effertz, P., Jaffe, D. A., Kubistin, D., Lynch, J., Petropavlovskikh, I., Puchalski, M., Sharac, T., Sive, B. C., Steinbacher, M., Torres, C., and Cooper, O. R. (2023), Fingerprints of the COVID-19 economic downturn and recovery on ozone anomalies at high-elevation sites in North America and western Europe, Atmos. Chem. Phys., 23, 15693–15709, https://doi.org/10.5194/acp-23-15693-2023

Szopa, S., V. Naik, B. Adhikary, P. Artaxo, T. Berntsen, W.D. Collins, S. Fuzzi, L. Gallardo, A. Kiendler-Scharr, Z. Klimont, H. Liao, N. Unger, and P. Zanis, 2021: Short-Lived Climate Forcers. In Climate Change 2021: The Physical Science Basis. Contribution of Working Group I to the Sixth Assessment Report of the Intergovernmental Panel on Climate Change [Masson-Delmotte, V., P. Zhai, A. Pirani, S.L. Connors, C. Péan, S. Berger, N. Caud, Y. Chen, L. Goldfarb, M.I. Gomis, M. Huang, K. Leitzell, E. Lonnoy, J.B.R. Matthews, T.K. Maycock, T. Waterfield, O. Yelekçi, R. Yu, and B. Zhou (eds.)]. Cambridge University Press, Cambridge, United Kingdom and New York, NY, USA, pp. 817–922, doi:10.1017/9781009157896.008.

---

## Author Comment (AC1)

**Author Responses to Reviewer Comments**

We thank the reviewers for their useful and constructive comments/feedback. We also thank Owen Cooper for his useful comments on our manuscript in relation to the TOAR-II special edition. We have reproduced their comments below in black text, followed by our responses in red text. Please note, where appropriate, we have number listed the reviewer comments for clarification. Any additions to the manuscript are in blue text and our reference to line numbers is based on the originally submitted manuscript.

**Reviewer #1's Comments:**

*Top Level Comments*

The manuscript entitled "Large Reductions in Satellite-Derived and Modelled European Lower Tropospheric Ozone During and After the COVID-19 Pandemic (2020–2022)" presents satellite records of European tropospheric $O_3$ from pre-COVID period to mid-2023 (post-COVID). In addition to providing evidence of $O_3$ reduction during the COVID-19 pandemic – consistent with findings from previous studies – the authors report an even larger reduction in $O_3$ in 2022, after the pandemic. This finding could hold scientific importance if further investigation into its underlying causes is conducted. Alongside the analysis of satellite products, the authors utilized a chemical transport model to separate the contributions of emissions and meteorology to $O_3$ reduction from 2020 to 2021. Their results suggest that meteorological factors, particularly stratosphere-troposphere exchange, play a crucial role in $O_3$ reduction. However, I find that the conclusions drawn in this section lack robustness due to the following reasons:

1. There is no $O_3$ budget analysis, which is necessary for a comprehensive evaluation of contributing factors.

Please see our response to Reviewer #1's General Comment #1.

2. A more thorough discussion of uncertainties related to the scaling factor, model resolution, and other relevant aspects is needed to enhance the scientific rigor of the study.

Please see our response to Reviewer #2's General Comment #2.

*General Comments*

1. Depth of scientific analysis: The current manuscript primarily presents data analysis and modeling results but does not sufficiently explore the underlying scientific implications. The discussion/conclusion sections are brief and lack a deeper interpretation of the findings. I recommend expanding these sections to provide more context, including possible explanations, comparisons with previous studies, and insights into the broader scientific relevance of the results. For example, the finding that the largest $O_3$ reduction occurred in 2022, after the COVID-19 period, could hold scientific significance if potential explanations for this trend were explored in greater depth.

In line with reviewer's comments, we have discussed the results in more detail (e.g. the Conclusions) and focussed the manuscript more on the key period (i.e. 2020 and 2021) when we have had lock downs and have the available emissions factors for the modelling. Therefore, we have weakened our focus on the year 2022 and updated the text and Abstract accordingly (see our response to Reviewer #2 Minor Comments #1). And while the reviewer suggests we run a BAU simulation in TOMCAT for 2022 (which  made sense given we emphasised this in the original Abstract), we believe that

with the refocussing of the abstract on 2020 and 2021, this would no longer provide extensive results which would greatly improve the manuscript. This is especially true since we don't have the scale factor data for 2022 to compare with the BAU case. Therefore, we politely refrain from undertaking this additional TOMCAT simulation.

Now, while we agree with much of Reviewer #1's General Comment #1, we politely disagree that our current work "does not sufficiently explore the underlying scientific implications". We have used two satellite products which both show a substantial decrease in lower tropospheric ozone, used the scale factor information from a prominent paper in Nature Climate Change (i.e. Forster et al., (2020)) and used a well evaluated model TOMCAT for this work. TOMCAT has been used for multiple peer-reviewed studies in Copernicus journals (e.g. Monks et al., (2017), Rowlinson et al., (2019), Richards et al., (2013), Pope et al., (2023) and Pope et al., (2024)). Therefore, we are confident in the tools we have used and the results we present here.

Here, the Conclusions have been updated to:

"Our study represents one of the first extended investigations of the COVID-19 pandemic impacts on European lower tropospheric $O_3$ (surface-450 hPa) using satellite observations and modelling. The records from the Global Ozone Monitoring Experiment (GOME – 2) and the Infrared Atmospheric Sounding Interferometer (IASI) on MetOp-B show substantial decreases in European average spring-summer time lower tropospheric ozone of typically 2.0-3.0 DU (or 11.0-14.6%). While not the key focus of this paper, the 2022 decline in $O_3$ is interestingly the largest between 2020 and 2023. Therefore, this would suggest other factors not investigated in this study are driving a more substantial $O_3$ decrease and that the reported COVID-19 response is within the more extreme variability of European ozone.

To investigate the drivers of the $O_3$ decreases over Europe during the COVID-19 period (2020-2021), activity scaling factors (i.e. based on anonymised mobility data from big tech firms) were used to perturb the model's business-as-usual (BAU) emissions for 2020 and 2021 to quantify the COVID-19 impact on $O_3$. Here, the TOMCAT simulations of lower tropospheric $O_3$ were reduced by 2.0-3.0 DU (comparable to the $O_3$ reductions reported by the satellite records) in the COVID-19 simulation compared to the BAU baseline. Further model sensitivity experiments were able to diagnose the contribution of 2020 emissions changes (approximately 66%) and 2020 meteorological conditions (approximately 34%) to the overall TOMCAT simulated $O_3$ reduction in Europe. Therefore, the COVID-19 reduced in $O_3$ precursor emissions were substantial in reducing 2020 European $O_3$, but it was amplified by meteorological conditions that year. Investigation of the TOMCAT stratospheric $O_3$ tagged tracer (i.e. a representation of the flux of stratospheric rich $O_3$ air into the troposphere) suggested a substantial drop in its contribution to lower tropospheric $O_3$ (in the order of 1.0 DU), which was comparable to the meteorological signal. Thus, a likely cause of the amplified European $O_3$ reduction in the COVID-19 period.

Therefore, our study has successfully quantified the impact of COVID-19 on European lower tropospheric ozone and identified a useful methodology to isolate the impact of emission changes, but also importantly meteorological variability, on observed changes in tropospheric composition. Future work would focus on the large reduction in European $O_3$ in 2022 (which is beyond the scope of this study), produce a harmonised IASI $O_3$ record from the three MetOp satellites and a reprocessing of the RAL Space GOME-2 record to more accurately account for UV-degradation in the instrument record.".

2. Uncertainty analysis and discussion: The analysis presented in this study generally lacks an assessment of uncertainty, which is crucial for interpreting and validating the results. Including uncertainty analysis in Section 3 and discussing limitations in the conclusion would enhance the robustness of the findings. Currently, the conclusion drawn from the modeling analysis is limited due to absence of uncertainty analysis. I recommend a more thorough discussion on uncertainties related to the scaling factor used, model resolution, etc. These additions would significantly improve the scientific rigor of the study.

We provide a response to Reviewer #1's Specific Comment 2 on the budgets of TOMCAT $O_3$ and have undertaken an assessment of the model's tropospheric $O_3$ burden to show it is a suitable tool for this study. In terms of the model horizontal resolution, Pope et al., (2023) undertook a detailed assessment of TOMCAT simulated over Europe (see Supplement of that study) finding it to be comparable with that of the higher resolution Copernicus Atmospheric Monitoring Service (CAMS) simulated ozone fields. In our response to Reviewer #1's Specific Comment 2 we have alluded to the skill of the model and suitability for studies such as this. We have also updated the Abstract (see our response to Reviewer #2's Minor Comments #1) and Conclusions (see our response to Reviewer #1's General Comments #1) in response to other reviewer comments. In terms of the scale factor used, this is based on mobility data from big tech companies like Apple and Google. Forster et al., (2020) found that the 3 or 4 datasets (depending on spatial region) of mobility data they had used were typically within 20% of each other with a correlation of 0.8 and higher. Therefore, these datasets where of sufficient quality for them to derive a "two-year blip" scenario for 2020 and 2021, which could be used to estimate changes in air pollutant and climate emissions, which was published in a high-profile paper. Therefore, given the past application of these datasets, we are confident in using the scale factors here. However, to make this clearer to the reader, this uncertainty information about the scale factor has been added to the manuscript and can be seen in our response to Reviewer #2's Minor Comment #3.

**Specific Comments**

1. Line 136: If the authors have done a literature review but still lack of an emission scaling factor for 2022, running a BAU case with 2022 meteorological data could provide valuable modeling insights into the causes behind the observed large $O_3$ reduction in 2022. This would significantly enhance the scientific impact of this study.

Please see our response to Reviewer #1's General Comment #1.

2. Line 156: The implementation of the STE tracer is somewhat difficult to understand. It seems unusual that the only chemical sinks, aside from photolysis, are reactions with $HO_2$/OH and $H_2O$ through $O(^1D)$ produced from the tracer itself. Could you provide more explanation on how this tracer could represents the $O_3$ budget or its contribution to tropospheric $O_3$? Alternatively, citing previous studies that use a similar tracer implementation could help, too.

The STE tracer has the same sinks in the troposphere as tropospheric ozone. There is the loss of the tracer and tropospheric ozone with e.g. NO but this is then recycled to reform ozone (i.e. the ozone-NOx relationship is a null cycle). Thus, only the final termination reactions have been listed. This STE scheme in TOMCAT has been used by several studies (e.g. Pope et al., (2023)), so we are confident in its application in this study.

In terms of budgets, unfortunately, TOMCAT does not output these diagnostics (e.g. ozone chemical production and sink terms) as standard for a complete budget analysis. It is beyond the scope of this

study to add these new diagnostics into the model although we agree this would be a useful addition to the model for future studies. However, we have calculated a tropospheric ozone budget for TOMCAT which is 322 Tg for 2020 in the BAU simulation and 314 Tg in the COVID-19 simulation. These values are consistent with that from Monks et al., (2017), who undertook a detailed assessment of the model. They found that TOMCAT simulated the tropospheric ozone burden to be 331 Tg, which sits within the ACCMIP range of 337±23 Tg (Young et al., 2013), as does our simulation here. Thus, providing confidence in TOMCAT and its application in this study. To make this clearer, we have updated the text on Page 5 Lines 160-162 to:

"Overall, TOMCAT is a robust and well evaluated CTM having been used in multiple studies of tropospheric $O_3$ and compared with many types of observation (e.g. Richards et al., (2013), Pope et al., (2020) and Pope et al., (2023). The simulated tropospheric ozone burden is a common metric to assess the skill of a model to simulate tropospheric ozone. Here, we derive a tropospheric $O_3$ burden of 322 Tg (BAU 2020 simulation), which is consistent with that of Monks et al., (2017) who reported an equivalent of 331 Tg. Both estimates sit within the reported range of 337±23 Tg from the Atmospheric Chemistry and Climate Model Intercomparison Project (ACCMIP, Young et al., 2013) further demonstrating TOMCAT to be a suitable modelling framework for this study.".

3. Fig2(a): In examining the sub-col $O_3$ time series from the two satellite products, I noticed that, in addition to a generally lower values across the observation period – which the authors suggest might be due to instrument biases – GOME-2B appears to have a one-month phase compared to the IASI data. For instance, GOME-2B consistently shows a lower trough at the end of 2015, 2016, 2017, 2018, and 2019, whereas IASI shows this trough at the beginning of each respective year. Could you offer some potential reasons for this discrepancy?

The European domain used in this study reaches nearly 70°N where sampling of GOME-2 is restricted in winter because of the high solar zenith angle. However, as IASI is an IR instrument, it is not impacted by the solar zenith angle. Therefore, this difference in sampling of the annual cycle at highest latitudes may be contributing a ~1-month offset in their domain average datasets.

We have discussed this in the manuscript Page 7 Line 195 as a new paragraph:

"It is worth noting that there is approximately a 1-month lag between the IASI and GOME-2 time-series in Figure 2 which is likely due to the European domain (see Figure S4 of the Supplement) extending to high northern latitudes (approximately 70°N) where sampling of the GOME-2 UV sounder, but not IASI, is restricted in winter months by absence of sunlight. While this could slightly influence the domain average annual cycle comparison it does not affect the interannual variability subject of this study.".

4. Line 223 (Figure 3(b)): Given that the represent an average of three simulations, I suggest adding error bars to indicate the standard deviation in the plot alongside the averaged values. This would help illustrate the significance of the differences more effectively.

While we agree that in principle it would be useful to add some estimate of the uncertainty or spread in the 3 model simulations, we do not consider 3 time-series to be a sufficient sample to calculate a standard deviation. Secondly, Figure 3b is already a busy plot, so adding more lines is likely to make it unclear. Finally, we are interested in the large-scale response to meteorology. We could have taken the meteorology for 2019 and run the 2020 simulation with 2019 meteorology alone. However, we thought it more sensible to use the average of 3 simulations for 2020 using the meteorology for 2017, 2018 and 2019, respectively. As year-to-year meteorology can be variable,

taking the 3-year average to get a mean meteorology baseline seems appropriate for this work. Thus, we propose to leave Figure 3b as it is.

5. Line 251: "These values represent the contribution of emission reduction and the corresponding contribution of meteorology" – do you mean that the ratio represents the contribution of emission reduction, while (1- ratio) represents the contribution of meteorology? Please consider rephrasing this sentence to improve clarity and readability.

Figure 4 represents the reduction of ozone in DU and the percentage contribution of emissions and meteorology to this ozone reduction. To make this clearer, we have updated Line 251 to: "These values represent the percentage contributions of the emission reductions (due to COVID-19) and meteorological conditions to the determined reduction in the lower tropospheric column zone.".

6. Line 260: As mentioned previously, the definition of STE is not very clear. Could you elaborate further on why it represents the contribution of $O_3$ from STE? Specifically, can the STE tracer defined in this study be transferred as contribution of $O_3$ from STE? For instance, Griffiths et al., 2020 validate their derived STE using O3 budget: STE = $LO_3+DO_3-PO_3$. Could you adopt a similar method to demonstrate that the STE used in this paper aligns with this $O_3$ budget equation? Additionally, I recommend including a detailed $O_3$ budget analysis to provide a more robust and concrete evaluation of contributing factors.

For the stratospheric $O_3$ tracer, TOMCAT does not have a budget metric hardwired into the model so deriving a new set of budget metrics (as in Griffiths et al., (2020)) is beyond the scope of this study. However, the tracer is a very useful tool to determine the contribution of stratospheric ozone to the tropospheric $O_3$ quantity on a grid box by grid box basis. For instance, Pope et al., (2023) used the stratospheric $O_3$ tracer to calculate the proportion of the summer 2018 ozone enhancement in the mid-troposphere (i.e. 500 mb) originated from the stratosphere. In the most extreme instance, approximately 40% of that mid-tropospheric ozone enhancement was coming from the stratosphere. Thus, we have implemented a similar approach here to see if stratospheric $O_3$ intrusion into the lower tropospheric column, or lack of, was contributing to the tropospheric $O_3$ signal over Europe in 2020. Overall, we have provided a description of the tracer and referenced two key papers (Monks et al., (2017) and Pope et al., (2023)) about the tracer and its application. Therefore, we argue that the use and description of the tracer is sufficiently covered in our manuscript.

7. Line 267: Concluding that other factors are likely neutral or even positive based solely on the STE tracer anomaly being larger than the $O_3$ anomaly may not be very robust. An $O_3$ budget analysis could provide a clearer understanding of the contributions from various factors, such as chemical production, chemical losses and deposition losses, all of which are influenced by meteorological parameters and affect $O_3$ concentrations.

Please see our response to Reviewer #1's Specific Comment #2.

**Reviewer #2's Comments:**

*Top Level Comments*

The paper analyzes significant reductions in European lower tropospheric ozone ($O_3$) levels during and after the COVID-19 pandemic (2020–2022), using satellite data and modeling. It attributes these reductions to decreased emissions from lockdown-related activity reductions, with the remainder influenced by meteorological factors. The study highlights persistent anomalies in spring and

summer ozone levels, with the largest reductions observed in 2022. I find the paper very interesting and suitable for publication after the following comments are taken into consideration:

***General Comments***

1. Meteorology is used as a broad term to explain and quantify its effect on the ozone reduction. In the section about the TOMCAT, no details about the met fields are provided. For example, I assume you use average tropospheric met fields? But what if you use near surface ones? Are transport/winds included? See a more specific comment hereafter.

TOMCAT has a model time-step of 30 minutes and uses 6-hourly meteorological reanalyses from ERA5 provided by ECMWF. These variables include the winds, temperature, pressure, relative humidity, cloud fields and mass fluxes etc. This is indicated on Page 5 Line 138. However, to make this clearer, we have reworded:

"TOMCAT is an off-line model driven by 6-hourly ERA-5 meteorological reanalyses (Hersbach et al., 2020)." to

"TOMCAT is an off-line model driven by 6-hourly ERA-5 meteorological reanalyses (e.g. temperature, relative humidity, winds; Hersbach et al., 2020), which are provided by the European Centre for Mid-Range Weather Forecasting (ECMWF). The ERA-5 meteorological reanalyses are provided on 137 vertical levels (surface to 1 hPa), which are interpolated onto the TOMCAT vertical grid (31 levels - see Monks et al., (2017) Figure 1).".

2. The sections about IASI/GOME is independent from the one with the simulation. The conclusion section does not attempt to link both neither. No mention of applying the AK to the model simulation so no comparison is attempted. Why is so?

TOMCAT has been extensively compared with both GOME-2 and IASI data sets from RAL Space in Pope et al., (2023). Here, we found the model to have decent skill when compared with both instruments. And while not yet published, a second study currently under review in ACP (Pimlott et al. - https://egusphere.copernicus.org/preprints/2024/egusphere-2024-3717/) investigates TOMCAT and RAL Space ozone datasets in depth. To highlight the skill of TOMCAT, used in multiple other studies, we have added the following text on Page 5 Line 144.

"TOMCAT has been used to investigate tropospheric ozone in multiple studies (e.g. Pimlott et al., 2022), Pope et al. (2023, 2024), Richards et al., (2013), Rowlinson et al., (2019)) and compared with a range of observations (e.g. surface observations (Monks et al., (2017), Richards et al., (2013)), ozonesonde data (e.g. Pope et al., (2024)) and satellite data (e.g. Pope et al., (2023)). The latter included a detailed comparison of lower tropospheric ozone between TOMCAT and GOME-2/IASI, where thorough consideration of the satellite averaging kernels (i.e. function of satellite vertical sensitivity when retrieving sub-column profiles of $O_3$) was taken in conjunction with the model, generally displaying good agreement  between them. Therefore, we are confident in using TOMCAT to directly investigate the impact of COVID-19 on lower tropospheric ozone over Europe."

3. It would have been nice to see a map of the decrease or a map of the decreasing trends to see if it is negative everywhere in Europe.

We have supplemented the analysis in sub-section 3.2 by including a new figure showing the difference between the TOMCAT COVID-19 simulation and the BAU scenario and also the difference between the TOMCAT COVID-19 simulation using average meteorology from 2017-2019 and the BAU scenario. This has now been added as the new Figure 4:

[Figure]

**Figure 4**: TOMCAT lower tropospheric ozone (DU) differences (March-May 2020 average) between a) the TOMCAT COVID and TOMCAT BAU simulations and b) the TOMCAT COVID simulation with 2017-2019 average meteorology (TOMCAT run for 2017, 2018 and 2019 with 2020 COVID emissions and the three simulations averaged together) and the TOMCAT BAU simulation.

The corresponding update to the text in sub-section 3.2 on Page 8 Lines 211-230:

"In 2020, scaling the emissions according to the mobility data estimates in Forster et al. (2020) (TOMCAT COVID scenario) caused a monthly reduction in tropospheric $O_3$ from March to December (**Figure 3(a)**). During January and February, the COVID and BAU scenarios are very similar, however, from March onwards the COVID scenario shows a negative difference compared to the BAU scenario, which peaks at 2.0 DU (8.3%) lower in May. This negative difference then reduces through the year to December (0.7 DU, 4.1%). **Figure 4a** shows the spatial impact of COVID-19 on lower troposphere ozone simulated by TOMCAT. The March-May 2020 average is typically 1.0-2.0 DU lower across the whole European domain. In 2021, the COVID scenario in Figure 3(a) shows consistent reductions in all months of the year, starting at 0.6 DU (3.4%) in January, peaking at 1.0 DU (4.3%) in May, and reducing towards the end of the year, ending with 0.6 DU (3.2%) in December. The temporal pattern of the reduction is similar to that in surface emissions (**Figure 1**), although with considerably smaller percentage decreases (peak of ~30% for surface emissions and ~8% for the resulting $O_3$ sub-column). This highlights the large emission reductions required for a sizeable reduction in European lower tropospheric $O_3$. To identify the impact of meteorology in 2020, the scaled emissions in 2020 were used in three separate simulations with the meteorology of 2017, 2018 and 2019, with an average of these three scaled emission simulations shown in **Figure 3(b)**. The 2020 COVID scenario record is broadly lower than the 2017/2018/2019 averaged scaled emission scenario, despite using the same surface emissions, which indicates that the meteorology of 2020 had a large impact on the tropospheric $O_3$ reduction. This is supported by **Figure 4b** which shows that across most of Europe, 2020 meteorological conditions where more conducive to lower tropospheric ozone loss (i.e. differences of -3.0 and -1.0 DU) than previous years. However, the domain average shown for March-May 2020 in **Figure 3b** is buffered by the positive differences (up to 1.0-1.5 DU) above 60°N. The impact of meteorology in 2020 is greatest in the spring-summer (**Figure 3b**), as the differences between these two timeseries is largest from February–July, peaking at a 1.1 DU difference in May. This demonstrates the importance of meteorology to the resulting $O_3$ in the spring-summer of 2020. The records are much more consistent from August to the end of the year, with absolute differences below 0.6 DU, indicating a reduced impact from meteorology in the second half of the year.".

The final figure has now been updated to Figure 5.

4. Not sure if the paper's data is included in TOAR but it would be interesting to use the module recommended by TOAR to calculate the trends shown in SI: https://gitlab.jsc.fz-juelich.de/esde/toar-public/toarstats.

Yes, we have submitted this to the TOAR-II special issue. However, we have not derived any long-term trends in this work requiring such methods. Granted, in Section 1 of the supporting material, we look at the time-series of GOME-2 but this is to correct the data. And for Figure 2 in the main manuscript, this is focusing on the step change in ozone due to COVID-19. So, we would politely suggest we do not need to use the TOAR statistical packages for trends in this paper.

***Specific Comments***

1. Was the RAL product validated? If so, please add a reference.

Yes, the GOME-2 product has been evaluated by Miles et al., (2015) while the IASI product has been evaluated Pimlott et al., (2022). To make this clear, we have added the following sentence on Page 4 Line 118:

"Here, the RAL Space GOME-2 and IASI-IMS retrieval schemes for lower tropospheric ozone have been independently evaluated against ozonesonde data in Miles et al., (2015) and Pimlott et al., (2022)."

We have also defined the term IMS on Page 4 Line 117 as "Infrared and Microwave Sounding (IMS)".

2. Line 126: "The MetOp-B record was adjusted according to monthly differences with the MetOp-A record in the overlap year of 2018". The community has always supposed that the two instruments are the same. If anything, IASI B is a reference radiance instrument because of its stability (and before that it was IASI A), how come there are differences? This is the first time I read about a possible difference between the two instruments' products.

There could be multiple reasons why there are subtle differences between IASI-A and IASI-B for lower tropospheric ozone. It is not uncommon for instruments, while using similar hardware and retrieval algorithms, to have differences (e.g. GOME vs. GOME-2) or systematic offsets. Unfortunately, it is beyond the scope of this study to investigate the causes of these subtle differences, however, we have documented them here, so if useful, the scientific community are aware of it. As such, we have removed this offset in our study to harmonise the IASI time-series to investigate the impact of COVID-19 on ozone.

3. Line 220. This phrase is confusing. You investigate the effect of meteorology in the following sentence and state here that it is emission dependent. Rephrase?

We are not totally sure of the issue here. Both emissions and meteorology will influence the negative ozone anomaly. In the paragraph containing Line 220, we are investigating the impact from *emissions*. We then start the next paragraph on Line 222 which discusses the impact of *meteorology* on the ozone anomaly. The impact of the emissions has been tested using the BAU and COVID emissions. The approach to test the impact of meteorology is then clearly outlined on Lines 222-224. Therefore, we argue that Line 222 is fine as it is.

4. Line 240. "Meteorology" is very broad. Does this include long range transport? What is the meteorological factor that is driving the year-to-year difference? It would be interesting to make a simple test on the sole influence of the temperature or cloud cover (or other?). Since temperature/photochemistry drives the O3 concentrations, you can check/validate if the

simulations with the pre COVID years meteorology are in line with the temperatures over Europe in 2020 vs 2017 to 2019 (even if from ERA5). IASI for example, has a L2 NRT temperature product that can be retrieved from EUMETSAT.

The term "meteorology" has been used by the authors before (e.g. Pope et al., 2023) as an umbrella term to cover temperature, humidity, pressure and winds (e.g. long-range transport). To make this clearer in the manuscript, we have added on Page 8 Line 224:

"Here, we use the term "meteorology" to represent meteorological variables such as temperature, pressure and humidity, but also the long-range transport (i.e. advection/convection) of air masses, which influence tropospheric chemistry.".

In terms of individual met variables, this is definitely an interesting point. Unfortunately, removing a single meteorological field in TOMCAT (e.g. temperature) and replacing it with another year's data is not practical. We were very interested in doing this for the study by Pope et al., (2023) but variables like temperature etc. are closely correlated with other variables (e.g. pressure, mass fluxes etc.), so introducing large step changes in the meteorological fields yields instabilities in the model. Pope et al., (2023) did try to look at the relationship between pressure and ozone during the 2018 summer heat wave ozone event. However, due to the non-linearities in the relationship, it was difficult to identify a direct correlation or link between the two variables (e.g. Figure 11 of that study).

5. Figure 3/STE discussion/Line 260: the discussion of the STE should come way earlier, before talking about Figure 4. Please rearrange or simply remove the STE from Figure 3.

In line with the reviewer's comment, we have split the paragraph starting on Page 8 Line 244 at Page 9 Line 248. The discussion of Figure 4 is then converted into its own separate paragraph. The paragraph on STE, starting on Page 9 Line 260, has been move up the manuscript to above discussion of Figure 4.

***Minor Comments:***

1. This abstract phrase is weird/not necessary because it is self-explanatory in the following phrase: "Rutherford Appleton Laboratory (RAL) retrieval products show large negative anomalies in the spring-summer periods of 2020–2022, with the largest in 2022, and smaller reductions in 2023."

In line with the reviewer's comment, and several others in the review process, we have updated the full Abstract to correct this sentence, make it clearer what the scaling factors for the emissions are and to emphasis our results focusing on the COVID-19 period, which is the main focus of our work:

"Activity restrictions during the COVID-19 pandemic caused large-scale reductions in ozone ($O_3$) precursor emissions, which in turn substantially reduced the abundance of tropospheric $O_3$ in the Northern Hemisphere. Satellite records of lower tropospheric column $O_3$ ($0 - 6$ km) from the Rutherford Appleton Laboratory (RAL) highlight these large reductions in $O_3$ during the COVID-19 period (2020), which persisted into 2021 and 2022. The European domain average $O_3$ reduction ranged between 2.0 and 3.0 Dobson units (DU) (11.0-14.6%). These satellite results were supported by the TOMCAT chemistry transport model (CTM) through several model sensitivity experiments to account for changes in emissions and impact of the meteorological conditions in 2020. Here, the business-as-usual (BAU) emissions were scaled by activity data (i.e. anonymised mobility data from big tech companies) to account for the reduction in $O_3$ precursor emissions. The model simulated large $O_3$ reductions (2.0-3.0 DU), similar to the satellite records, where approximately 66% and 34%

of the $O_3$ loss can be explained by emissions changes and meteorological conditions, respectively. Our results also show that the reduced flux of stratospheric $O_3$ into the troposphere accounted for a substantial component of the meteorological signal in the overall lower tropospheric $O_3$ levels during the COVID-19 period.".

2. The first phrase of the text should read "tropospheric ozone ($O_3$)"

This has been corrected.

3. L60: "Based on activity data », what does this even mean?

The data used by Forster et al., (2020) was based on Google and Apple mobility data (see references within). Here, the activity data is based on changes in people's activities which was sampled via methods such as mobile GPS etc. The data provided by Google and Apple has been anonymised. Therefore, to make this clearer, we have added the following text on Line 62:

"Here, the changes in activity data reported by Forster et al., (2020) are based on changes in anonymised mobility data (e.g. from phone GPS information) provided by Apple and Google (see Forster et al., (2020) and references within). Typically, they found these mobility datasets used in their study to be within 20% of each other and had a correlation of 0.8 or higher.".

4. MetOp is, since few years, it spelled Metop (no capital O)

We believe that MetOp has been interchangeable spelt as MetOp and Metop depending on the study or website. Therefore, to be consistent with our past studies, we propose to keep using MetOp.

5. L160-162. This phrase should come earlier.

This sentence has now been removed. In response to the Reviewer #2's General Comment #2, we have now added an equivalent piece of text, but more detailed, further up in the paragraph on Line 144.

6. Figure 3. Spell STE in the legend

To make this clearer, we have now spelled out what STE in the Figure 3 caption.

**Owen Cooper's Comments:**

*Top Level Comments*

1. Comments regarding TOAR-II guidelines: TOAR-II has produced two guidance documents to help authors develop their manuscripts so that results can be consistently compared across the wide range of studies that will be written for the TOARII Community Special Issue. Both guidance documents can be found on the TOAR-II webpage: https://igacproject.org/activities/TOAR/TOAR-II

Thank you for making us aware of these resources. We have tried to adhere to these in this manuscript and others we have submitted to TOAR-II wherever possible.

2. The TOAR-II Community Special Issue Guidelines: In the spirit of collaboration and to allow TOAR-II findings to be directly comparable across publications, the TOAR-II Steering Committee has issued this set of guidelines regarding style, units, plotting scales, regional and tropospheric column comparisons, and tropopause definitions.

3. The TOAR-II Recommendations for Statistical Analyses: The aim of this guidance note is to provide recommendations on best statistical practices and to ensure consistent communication of statistical analysis and associated uncertainty across TOAR publications. The scope includes approaches for reporting trends, a discussion of strengths and weaknesses of commonly used techniques, and calibrated language for the communication of uncertainty. Table 3 of the TOAR-II statistical guidelines provides calibrated language for describing trends and uncertainty, similar to the approach of IPCC, which allows trends to be discussed without having to use the problematic expression, "statistically significant".

Thank you for making us aware of these resources. We have tried to adhere to these in this manuscript and others we have submitted to TOAR-II wherever possible.

***General Comments***

1. A new paper published in the TOAR-II Community Special Issue is highly relevant to this study as it shows that the decrease of ozone observed in the free troposphere during 2020 also extended to the surface, as observed at high elevation monitoring sites in North America and Europe (Putero et al., 2023).

We have added the Putero et al., (2023) study to our literature review in the Introduction.

2. An additional relevant study: Every year the State of the Climate reports provide updates on the global distribution and trends of greenhouse gases, including tropospheric ozone. The most recent report (Dunn et al., 2024) reports the latest findings based on NASA's OMI/MLS tropospheric ozone product (see Figure 2.66 on page S95). The data show an increase of the tropospheric ozone burden (60° S – 60° N) from 2004 to 2019, followed by a drop in ozone in 2020 and a levelling off through 2023.

We have added the Dunn et al., (2024) report to our literature review in the introduction.

3. Lines 76-78: The authors cite a paper that claims that free tropospheric ozone is decreasing all across northern midlatitudes, but no other study has been able to replicate those results. In contrast, plenty of studies have conducted in-depth analysis of ozone observations in the free troposphere, and do not find a decrease of ozone. IPCC AR6 (Gulev et al., 2021; Szopa et al., 2021) concluded that free tropospheric ozone has increased since the mid-1990s based, in part, on IAGOS observations in the free troposphere (Gaudel et al., 2020). And follow-up studies have shown that ozone increased in the free troposphere from the mid1990s to 2019 above Europe and western North America, with a decrease of ozone in 2020 due to the COVID-19 pandemic (Chang et al., 2022, 2023). These in situ observations fit with the decrease of ozone in 2020 and 2021, as observed by several satellite products (Ziemke et al., 2022; Dunn et al., 2024). Another recent study, submitted to the TOAR-II Community Special Issue, uses the ECHAM6-HAMMOZ global atmospheric chemistry model to show that ozone generally increased across the northern hemisphere from 2004 to 2019, in agreement with the OMI/MLS satellite product (see Figure 4 of Fadnavis et al., 2024). A NASA study reached similar conclusions (Liu et al., 2022).

For the comment "The authors cite a paper that claims that free tropospheric ozone is decreasing", we agree that most studies find either a plateau (e.g. North America and Europe) or an increase

(East Asia) in the northern mid-latitudes. However, the study in question Parrish et al., (2022) is published in ACP and authors/authors have a substantial history of research in this area. So, we are inclined to keep this reference in our manuscript. We also make it clear on Line 76 that the Parrish study is different by saying "In contrast,".

We also already cite several of the other papers mentioned (i.e. Ziemke et al., (2022) and Chang et al., (2022). We have also added the Dunn et al., (2024) reference in our response to Owen Cooper's General Comment #2. The Fadnavis et al., paper is an interesting study (several of the authors here are co-authors on that paper) but it is still under review, so we cannot reference it. Therefore, hopefully we have added sufficient citations to improve our literature review in the Introduction.

**Minor Comments**

1. Line 50 missing the word "of"?

This has been corrected.

2. Lline 293 missing the word "ozone"

This has been corrected.

**References**:

Dunn, R. J. H., J. Blannin, N. Gobron, J. B Miller, and K. M. Willett, Eds., 2024: Global Climate [in "State of the Climate in 2023"]. Bull. Amer. Meteor. Soc., 105 (8), S12–S155, https://doi.org/10.1175/BAMS-D24-0116.1.

Pope, R. J., Arnold, S. R., Chipperfield, M. P., Reddington, C. L. S., Butt, E. W., Keslake, T. D., et al. (2020). Substantial increases in Eastern Amazon and Cerrado biomass burning-sourced tropospheric ozone. *Geophysical Research Letters*, 47, e2019GL084143. https://doi.org/10.1029/2019GL084143.

Pope, R. J., Rap, A., Pimlott, M. A., Barret, B., Le Flochmoen, E., Kerridge, B. J., Siddans, R., Latter, B. G., Ventress, L. J., Boynard, A., Retscher, C., Feng, W., Rigby, R., Dhomse, S. S., Wespes, C., and Chipperfield, M. P.: Quantifying the tropospheric ozone radiative effect and its temporal evolution in the satellite era, Atmos. Chem. Phys., 24, 3613–3626, https://doi.org/10.5194/acp-24-3613-2024, 2024.

Putero, D., Cristofanelli, P., Chang, K.-L., Dufour, G., Beachley, G., Couret, C., Effertz, P., Jaffe, D. A., Kubistin, D., Lynch, J., Petropavlovskikh, I., Puchalski, M., Sharac, T., Sive, B. C., Steinbacher, M., Torres, C., and Cooper, O. R.: Fingerprints of the COVID-19 economic downturn and recovery on ozone anomalies at high-elevation sites in North America and western Europe, Atmos. Chem. Phys., 23, 15693–15709, https://doi.org/10.5194/acp-23-15693-2023, 2023.

Rowlinson, M. J., Rap, A., Arnold, S. R., Pope, R. J., Chipperfield, M. P., McNorton, J., Forster, P., Gordon, H., Pringle, K. J., Feng, W., Kerridge, B. J., Latter, B. L., and Siddans, R.: Impact of El Niño–Southern Oscillation on the interannual variability of methane and tropospheric ozone, Atmos. Chem. Phys., 19, 8669–8686, https://doi.org/10.5194/acp-19-8669-2019, 2019.

Young, P. J., Archibald, A. T., Bowman, K. W., Lamarque, J.-F., Naik, V., Stevenson, D. S., Tilmes, S., Voulgarakis, A., Wild, O., Bergmann, D., Cameron-Smith, P., Cionni, I., Collins, W. J., Dalsøren, S. B., Doherty, R. M., Eyring, V., Faluvegi, G., Horowitz, L. W., Josse, B., Lee, Y. H., MacKenzie, I. A., Nagashima, T., Plummer, D. A., Righi, M., Rumbold, S. T., Skeie, R. B., Shindell, D. T., Strode, S. A., Sudo, K., Szopa, S., and Zeng, G.: Pre-industrial to end 21st century projections of tropospheric ozone

from the Atmospheric Chemistry and Climate Model Intercomparison Project (ACCMIP), Atmos. Chem. Phys., 13, 2063–2090, https://doi.org/10.5194/acp-13-2063-2013, 2013.